# Repeated Stereotactic Radiotherapy for Local Brain Metastases Failure or Distant Brain Recurrent: A Retrospective Study of 184 Patients

**DOI:** 10.3390/cancers15204948

**Published:** 2023-10-11

**Authors:** Laure Kuntz, Clara Le Fèvre, Delphine Jarnet, Audrey Keller, Philippe Meyer, Christophe Mazzara, Hélène Cebula, Georges Noel, Delphine Antoni

**Affiliations:** 1Department of Radiation Oncology, Institut de Cancérologie Strasbourg Europe (ICANS), UNICANCER, Paul Strauss Comprehensive Cancer Center, 17 Rue Albert Calmette, 67200 Strasbourg, France; l.kuntz@icans.eu (L.K.); c.lefevre@icans.eu (C.L.F.); a.keller@icans.eu (A.K.); d.antoni@icans.eu (D.A.); 2Department of Medical Physics, Institut de Cancérologie Strasbourg Europe (ICANS), UNICANCER, Paul Strauss Comprehensive Cancer Center, 17 Rue Albert Calmette, 67200 Strasbourg, France; d.jarnet@icans.eu (D.J.); p.meyer@icans.eu (P.M.); c.mazzara@icans.eu (C.M.); 3Neurosurgery Department, Hôpitaux Universitaires de Strasbourg, 1, Avenue Molière, 67098 Strasbourg, France; helene.cebula@chru-strasbourg.fr

**Keywords:** radiotherapy, salvage radiation, stereotactic radiosurgery, brain metastases, reirradiation, repeated radiosurgery, oligorecurrence

## Abstract

**Simple Summary:**

Brain metastases are the most common intracranial malignant neoplasms in adults. Stereotactic radiotherapy (SRT) is one of the most widely used therapeutic modalities to treat BMs. The main benefit of SRT is to delay whole-brain radiotherapy, which causes cognitive impairment. About 20–40% of patients will require salvage treatment after an initial SRT session because of local failure or due to distant failure. The aim of this retrospective monocentric study was to evaluate the factors affecting overall survival (OS) and neurological death-free survival (NDFS). Patients receiving multiple repeated SRT for locally recurrent brain metastases or distant brain failure have better OS and NDFS than those receiving only two SRT sessions. The patients who would benefit most from repeat SRT are those treated for kidney or breast cancer, those in good general condition, those who did not receive systemic treatment and without extracerebral metastases and those with a low brain metastasis velocity.

**Abstract:**

Background: The main advantages of stereotactic radiotherapy (SRT) are to delay whole-brain radiotherapy (WBRT) and to deliver ablative doses. Despite this efficacy, the risk of distant brain metastases (BM) one year after SRT ranges from 26% to 77% and 20 to 40% of patients required salvage treatment. The role and consequences of reirradiation remain unclear, particularly in terms of survival. The objective was to study overall survival (OS) and neurological death-free survival (NDFS) and to specify the prognostic factors of long-term survival. Methods: we retrospectively reviewed the data of patients treated between 2010 and 2020 with at least two courses of SRT without previous WBRT. Results: In total, 184 patients were treated for 915 BMs with two-to-six SRT sessions. Additional SRT sessions were provided for local (5.6%) or distant (94.4%) BM recurrence. The median number of BMs treated per SRT was one with a median of four BMs in total. The mean time between the two SRT sessions was 8.9 months (95%CI 7.7–10.1) and there was no significant difference in the delay between the two sessions. The 6-, 12- and 24-month NDFS rates were 97%, 82% and 52%, respectively. The 6-, 12- and 24-month OS rates were 91%, 70% and 38%, respectively. OS was statistically related to the number of SRT sessions (HR = 0.48; *p* < 0.01), recursive partitioning analysis (HR = 1.84; *p* = 0.01), salvage WBRT (HR = 0.48; *p* = 0.01) and brain metastasis velocity (high: HR = 13.83; *p* < 0.01; intermediate: HR = 4.93; *p* < 0.01). Conclusions: Lung cancer and melanoma were associated with a lower NDFS compared to breast cancer. A low KPS, a low number of SRT sessions, synchronous extracerebral metastases, synchronous BMs, extracerebral progression at SRT1, a high BMV grade, no WBRT and local recurrence were also associated with a lower NDFS. A high KPS at SRT1 and low BMV grade are prognostic factors for better OS, regardless of the number of BM recurrence events.

## 1. Introduction

Brain metastases (BMs) are the most common intracranial malignant neoplasms in adults [1] and 10–40% of oncology patients will develop BMs during their oncologic history. BMs are caused by lung cancer in 40–50% of cases, followed by breast cancer (15–30%), melanoma (10%), colorectal cancer (3–8%) and kidney cancer (2–4%) [2]. With the efficiency of systemic therapies, the advent of targeted therapies, the increasing life expectancy of patients receiving these treatments and, finally, the growing elderly population, patients with a personal history of cancer are at higher risk of developing BMs and recurrence.

From several trials involving brain metastasis patients, irrespective of the treatment modality, the one-year local recurrence rate of already irradiated BM ranges from 5% to 38% and the one-year distant cerebral recurrence rate ranges from 22 to 67% [3,4,5,6,7]. The treatment of these new BM events is based on the same indication for the first event and includes stereotactic radiotherapy (SRT) [8]. The main advantage of SRT is to delay WBRT, which is implicated in the development of cognitive impairments [9,10]. The second advantage of SRT is to deliver ablative doses with a strong dose gradient, allowing for an increase in the local control rate [11,12]. Despite this efficacy, the risk of distant BM one year after SRT alone ranges from 26% to 77% [3,13]. In a systematic review, Kuntz et al. showed that 20 to 40% of patients required salvage treatment after the initial SRT session [14]. However, the role and consequences of reirradiation remain unclear, particularly in terms of survival.

Our monocentric study reports on a retrospective analysis of patients who underwent repeated SRT sessions without intercurrent WBRT. The objective of this analysis was to study overall survival (OS) and neurological death-free survival (NDFS) and to specify the prognostic factors of long-term survival.

## 2. Materials and Methods

### 2.1. Patients and Treatment Modalities

We queried our institutional database to obtain a list of patients who underwent two or more SRT sessions for cerebral or local recurrence for single or multiple BMs by session. Patients who had received whole-brain radiotherapy (WBRT) before an SRT or WBRT after only one SRT session were excluded. We identified 184 patients treated between January 2010 and June 2020. A total dose of 33 Gy was prescribed to the isocenter, administered in 3 fractions of 11 Gy every two days. Additionally, a one-fraction protocol, delivering 20 Gy to the isocenter, was preferred when the lesion measured less than 1 cm. The treatment plans were designed such that the 70% isodose line encompassed the Planning Target Volume (PTV), corresponding to doses of 23.1 Gy or 14 Gy, respectively. [15]. Treatments were delivered using dynamic conformational arcs of volumetric modulated arc therapy (VMAT). Stereotactic planning computed tomography and planning MRI scans were imported to iPlan RT Image (Brainlab^®^ AG, Feldkirchen, Germany) or Somavision (Varian Medical Systems^®^, Palo Alto, CA, USA) for image registration, fusion, contouring and planning.

### 2.2. Follow-Up

MRI was performed just before the follow-up consultation every three to six months and more frequently, if appropriate, depending on the neurological symptoms. OS was defined as the time between the diagnosis of the first BM and the date of death or last contact. Death from any cause was recorded, as well as death from a neurologic cause. When the patient passed away at home or in another healthcare facility, the cause of death was predominantly recorded as unknown. In cases where the patient passed away in the hospital, death from a neurological cause was defined as either status epilepticus or brain involvement.

### 2.3. Brain Metastasis Velocity

The final BMV was calculated for each patient according to Farris et al., similar to Yamamoto et al., who confirmed the validity of BMV in predicting OS after the second SRT session, but also after the third and fourth session [16,17]. BMV was classified into low-, intermediate- or high-risk groups if the number of new BMs was <4, 4 to 13 and >13, respectively. The BMV grade was calculated after the second, the third and the fourth stereotactic radiotherapy session following the same methodology, SRT2, SRT3 and SRT4, respectively.

### 2.4. Statistical Analysis

Statistical analysis was performed using R Studio software version 1.3.1093 (© 2009–2020 RStudio, PBC) using the packages “survival”, “prettyR” and “gmrcfun”. Quantitative variables and qualitative data are expressed using standard position and dispersion statistics and proportions. Univariate analysis was performed using Pearson’s linear correlation test, Spearman correlation test, Kruskal and Wallis test, parametric Chi2 test or Fisher test if the conditions of application allowed them. Multivariate analysis was performed using logistic regression tests. Survival analyses were performed using the Cox model. The alpha risk was set at 5% for all analyses.

## 3. Results

### 3.1. Patient Characteristics

A total of 184 patients were enrolled; the median age at the diagnosis of cancer was 58 years old and that at the diagnosis of BMs was 61 years old. Fifty-six percent of the patients were followed for lung cancer, 13% for breast cancer (among them, 45.8% were HER2-positive), 13% for melanoma, 8.7% for cancers from the digestive tract and 4.4% for kidney cancer. At the first SRT session (SRT1), the initial primary tumor was controlled in 59.2% of the patients and synchronic extracerebral metastases (ECMs) were diagnosed in 63.3%. Table 1A shows the initial characteristics of the patients and Table 1B shows the characteristics of the patients at each SRT session. The further analyses of changes in patient characteristics over time are available in a dedicated article [18]. Repeated SRT does not lead to the frank alteration of the general condition of the patients and KPS was maintained over 70% for more than 95% of our patients during all SRT sessions.

One hundred and three patients (66.8%) received a total of two sessions of SRT; SRT1 was delivered at the time of the treatment of the first BMs and SRT2 was delivered at the time of the treatment of the locally recurrent BM or distant brain failure. Thirty-nine patients (21.2%) underwent two locally recurrent BMs or distant brain failure and received three SRT sessions. Fourteen patients (7.6%) underwent three locally recurrent BMs or distant brain failure and received four SRT sessions. Seven patients (3.8%) underwent four locally recurrent BMs or distant brain failure and received five SRT sessions. Additionally, one patient (0.5%) underwent five locally recurrent BMs or distant brain failure and received six SRT sessions. Figure 1 shows the patient outcomes according to the BMV grade starting from SRT2.

### 3.2. Brain Metastases Characteristics

Nine hundred and fifteen BMs were treated for a total of 460 SRT sessions. Each patient had a mean of two BMs treated per SRT session (range: 1–6; 95% CI 1.84–2.16), for a total average of five BMs treated during all sessions (range: 2–19; 95% CI 4.52–5.44). The median number of BMs was 1.5 (95% CI 1.0–6.0) at SRT1, 1 (95% CI 1.0–5.0) at SRT2, 1 (95% CI 1.0–4.0) at SRT3 and 1 (95% CI 1.4–2.2) at SRT4, etc. There was no significant difference in the number of BMs between the consecutive sessions of SRT (*p* = 0.06, *p* = 0.48 and *p* = 0.8, respectively). The number of BMs treated per SRT session was not significantly different between sessions, but there was a tendency for there to be more BMs in SRT1 (*p* = 0.06). The median GTVs of each BM at SRT1, SRT2, SRT3 and SRT4, etc., were 0.4 mL (95% CI 1.84–2.94), 0.4 mL (95% CI 2.67–4.77), 0.25 mL (95% CI 1.42–4.43) and 0.35 mL (95% CI 1.16–4.16), respectively. The median total GTVs per session (i.e., the sum of the GTVs treated during an SRT session) were 6.2 mL (95% CI mean 7.01–9.9), 0.9 mL (95% CI mean 3.3–6.8), 0.6 mL (95% CI mean 1.3–4.8) and 1 mL (95% CI mean 0.1–10.2) at SRT1, SRT2, SRT3, SRT4 and thereafter, respectively. The total GTVs at SRT1 were statistically higher than the GTVs at SRT2 (*p* < 0.001), but this difference was not found between SRT2 and SRT3 (*p* = 0.34) or between SRT3 and SRT4, etc., (*p* = 0.62). A high total GTV at SRT1 was not statistically associated with synchronous brain metastases at diagnosis (*p* = 0.43).

### 3.3. Time between Each Session

The median time between each session, regardless of the number of SRT sessions, was 6.7 months (95% CI 7.8–10.1). The median time between SRT1 and SRT2 was 6.6 months (95% CI 7.9–10.6) and 5.3 months (95% CI 6.1–11.5) between SRT2 and SRT3, 7.8 months (95% CI 5.3–15.1) between SRT3 and SRT4 and 7.2 months (95% CI 0.6–28.1) between SRT4–5 and SRT5–6. There was no statistically significant difference between the SRT1-SRT2 and SRT2-SRT3 time, between SRT2-SRT3 and SRT3-SRT4 or between SRT3-SRT4 and SRT4-SRT5 (*p* = 0.19, *p* = 0.89, *p* = 0.17, respectively).

For low, intermediate and high BMV grades, the median times between the first and the second stereotactic radiotherapy were 10.2 months (95% CI 11.0–15.2), 3.0 months (95% CI 3.7–4.9) and 2.5 months (95% CI 2.0–3.0), respectively (*p* < 0.001). The median times between SRT2 and SRT3 were 6.0 months (95% CI 7.1–14.8), 5.0 months (95% CI 3.7–5.9) and 2.6 months (95% CI 0.3–4.1) for low, intermediate and high BMV grades, respectively (*p* < 0.001). The median times between SRT3 and SRT4 were 8.5 months and 7.3 months for low and intermediate BMV grades, respectively (*p* = 0.22).

### 3.4. Overall Survival

The median follow-up time of the whole population was 18.4 months (range: 2–95). At this time point, 20.1% patients were still alive. Among the 79.9% patients who died, 28% died of neurological causes. The median OS was 18.6 months (95% CI 17.0–21.1). The 6-, 12- and 24-month OS rates were 91% (95% CI 88–96), 70% (95% CI 64–77) and 38% (95% CI 32–45), respectively. In the univariate analysis, the OS rates were related to sex (*p* < 0.01), history of hypertension (*p* = 0.05), diabetes (*p* = 0.01), the location of the primary tumor (*p* = 0.02), the tumor stage (*p* = 0.02), the number of SRTs (*p* < 0.01), ECOG at SRT1 (*p* < 0.01), KPS at SRT1 (*p* < 0.01), RPA at SRT1 (*p* = 0.01), the extracerebral progression status at SRT1 (*p* < 0.01), systemic treatment at SRT1 (*p* = 0.04), the DS-GPA score (taking into account the lowest DS-GPA score of all the SRT sessions for each patient; *p* < 0.01), the RPA score (taking into account the highest RPA score of all the SRT sessions for each patient; *p* = 0.01), targeted therapy (*p* = 0.02), the BMV grade (*p* < 0.01), salvage WBRT (*p* = 0.04), local recurrence (*p* < 0.01) and radionecrosis (*p* < 0.01).

In the multivariate analysis, the primary cancer (melanoma versus breast cancer: HR = 8.6; 95% CI 2.3–31.8; *p* < 0.01), the number of SRTs ≥ 3 (HR = 0.24; 95%CI 0.12–0.47; *p* < 0.01), a high KPS at SRT1 (HR = 0.96; CI95% 0.95–0.98; *p* < 0.01), no systemic treatment at SRT1 (HR = 2.48; 95%CI 1.2–5.1; *p* = 0.01), the BMV grade (low versus high: HR = 125.1; 95% CI 30.8–508.1; *p* < 0.01 and low versus intermediate: HR = 3.7; 95%CI 1.9–7.0; *p* < 0.01) and no salvage WBRT (HR = 0.49; 95%CI 0.24–0.99; *p* = 0.01) were favorable prognostic factors of OS. Figure 2 shows the OS according to the BMV grade and total number of SRT sessions.

The 6-, 12- and 24-month NDFS rates were 97%, 82% and 52%, respectively. In the univariate analysis, NDFS was related to the primary cancer (*p* = 0.04), the tumor stage (*p* = 0.03), the number of SRTs ≥ 3 (*p* = 0.03), the number of BMs ≥ 10 (*p* = 0.03), the total cumulative metastasis volume ≥ 3 mL (*p* = 0.03) and the BMV grade (*p* < 0.01). In the multivariate analysis, breast cancer (digestive versus breast cancer: HR = 15.0, 95%CI 4.1–55.4, *p* < 0.001; lung versus breast cancer: HR = 9.64, 95% CI 4.0623.3, *p* < 0.001; melanoma versus breast cancer: HR = 5.59, 95% CI 1.7–18.1, *p* < 0.01; other cancers versus breast cancer: HR = 96.07, 95% CI 11.7–790.0, *p* < 0.001), the early tumoral stage (HR= 2.89, 95% CI 1.8–4.6, *p*< 0.001), no brain metastasis at the primary cancer diagnostic (HR = 0.31, 95% CI 0.1–0.9, *p* = 0.01), the initial absence of ECM (HR = 0.17, 95% CI 0.07–0.44, *p* < 0.01), the number of SRTs ≥ 3 (HR = 0.28, 95% CI 0.1–0.8, *p* = 0.02), a high KPS at SRT1 (HR = 0.97, 95% CI 0.95–0.99, *p* = 0.02), no extracerebral progression at SRT1 (HR = 7.88, 95%CI 2.9–21.3, *p* < 0.01), the BMV grade (low versus high: HR = 851.5; 95% CI 85.6–8468.7, *p* < 0.001; and low versus intermediate: HR = 22.2; 95% CI 7.7–63.8, *p* < 0.01), no salvage WBRT (HR = 0.049; 95% CI 0.01–0.23, *p* < 0.001), no local recurrence (HR = 3.56; 95% CI 1.5–8.7, *p* < 0.01) and the presence of radionecrosis (HR = 0.11, 95% CI 0.04–0.33, *p* < 0.01) were favorable prognostic factors of NDFS. The prognostic factors for OS and NDFS are summarized in Table 2. Appendix A (available in Appendix A) show OS and NDFS in function of main prognostic factors, respectively.

### 3.5. Salvage Whole-Brain Radiotherapy (WBRT)

Thirty-four (18.5%) patients ultimately underwent salvage WBRT an average of 4.3 months after the last SRT session. Salvage WBRT was performed after SRT2, SRT3 or SRT4 in 30, 3 and 1 patients, respectively (mean: 2; min-max: 2–5). Patients who underwent WBRT had a mean number of five (min-max: 1–20) evolutive BMs compared to one (min-max: 1–8) for patients who received a third SRT session (*p* < 0.001). Finally, the high, intermediate and low BMV grade distributions were significantly different in patients in the WBRT group, with 5.9%, 55.9% and 38.2%, respectively, compared to 6.7%, 33.3% and 60%, respectively, in the WBRT-free group (*p* = 0.04). Additional statistical analyses of brain recurrence and local recurrence found no association with the primary cancer [19].

Patients treated with WBRT were significantly more likely to die (*p* = 0.01 in multivariate analysis), with death occurring at an average of 8.8 months (95% CI 5.9–11.7) after WBRT. When we separated the patients into three groups, namely, patients who had received WBRT, patients who were treated with two sessions of SRT without WBRT and patients who were treated with three or more sessions without WBRT, there was a statistically significant difference in OS among the three groups (*p* < 0.001). The median survival of WBRT patients and two-SRT-WBRT-free patients was 17.3 months (95% CI 16.2–21.1) and 13.8 months (95% CI 10.9–17.1), respectively (*p* = 0.89), which suggests that there is a trend toward a longer OS after performing WBRT than when only supportive care is provided. Among the 93 patients who underwent only two SRT sessions without WBRT, 15% were still alive at the end of the follow-up. Neither the number of BMs (*p* = 0.21) nor systemic treatment (*p* = 0.37) was associated with WBRT. A high BMV grade and the presence of NDFS were statistically associated with WBRT, with *p* = 0.04 and < 0.01, respectively. Figure 3 shows the OS curves of patients according to the salvage treatment.

### 3.6. Outcome of Long-Surviving Patients

Among all patients, 37.5% survived more than two years after the first session of SRT and 18.5% were still alive at the end date of the study. The median survival time of the long-term survivor group after SRT1 was 43.0 months (95% CI 43.6–53.1). The median number of total SRT sessions was three (95% CI 2.7–3.2) versus two (95% 2.1–2.3) in the short-term survivor group (*p* < 0.01). The number of total BMs was not significantly different between long- and short-term survivors, with a median of four (*p* = 0.15). The time between the SRT sessions was statistically longer for long-term survivors, with a median time of 11.3 months (95% CI 12.1–18.2) between SRT1 and SRT2 versus 5.2 months (95% CI 5.0–6.4) for short-term survivors (*p* < 0.01). Furthermore, 1 (1.4%) long-term survivor had a high BMV grade, 12 (17.4%) had an intermediate BMV grade and 56 (81.2%) had a low BMV grade versus 11 (9.6%), 57 (49.5%) and 47 (40.9%) short-term survivors, respectively (*p* < 0.01). WBRT was performed as a salvage treatment in seven (10.1%) long-term survivors compared to 27 (23.5%) short-term survivors (*p* = 0.04).

### 3.7. Overall Survival Based on the Number of SRT Sessions (OSSRT)

We analyzed OS after SRT1 and SRT3 by taking the time from SRT1 to death as the survival interval for patients who underwent fewer than three SRT sessions and the time from SRT3 to death for patients who underwent three or more SRT sessions, respectively. The aim of this analysis was to highlight the initial and early characteristics of long-surviving patients after three SRT sessions. The median OSSRT was 15.7 months (95% CI 11.9–17.3) after SRT1 for patients treated with two sessions and 14.9 months (95% CI 8.7–38.5) after SRT3 for patients treated with three or more sessions (*p* = 0.6). The median NDFS was 19 months (95% CI 16.8-NA) after SRT1 for patients treated with two sessions and 13 months (95% CI 10.6-NA) for patients treated with three or more sessions (*p* = 0.1). Univariate and multivariate analyses of the OSSRT interval were performed based on the patient characteristics at cancer diagnosis and at SRT1/SRT2 and the results are shown in Table 3. In the univariate analysis, the age at the initial cancer diagnosis (*p* = 0.007), the initial tumoral stage (*p* = 0.03), ECOG at SRT1/2 (*p* < 0.001 and *p* < 0.001), KPS at SRT1/2 (*p* < 0.001 and *p* < 0.001), RPA at SRT1/2 (*p* = 0.004 and *p* < 0.001), extracerebral progression at SRT1/2 (*p* = 0.007 and *p* = 0.05), systemic treatment at SRT1/2 (*p* = 0.01 and *p* = 0.008) and the BMV grade (*p* < 0.001) were statistically associated with a better OSSRT. In the multivariate analysis, only a high KPS at SRT1 (HR = 0.93, 95% CI 0.87–0.99, *p* = 0.03) and a low BMV grade at SRT2 (low versus high: HR= 4.8, 95% CI 1.3–18.1, *p* = 0.02; low versus intermediate: HR = 2.0, 95% CI 1.04–3.7, *p* = 0.04) were statistically associated with a better OSSRT.

## 4. Discussion

To our knowledge, no other retrospective study has examined as many patients and BMs treated with consecutive SRT for relapse or new BMs. In this retrospective, monocentric study, patients receiving repeated SRT for local or cerebral recurrent BMs without WBRT were studied. A high number of SRT sessions, a high KPS at SRT1, no systemic treatment at SRT1, no WBRT and a low BMV grade were associated with improved OS in the multivariate analysis. Lung cancer and melanoma were associated with a lower NDFS compared to breast cancer. Low KPS, a high tumoral stage at the diagnosis of the primary cancer, a low number of SRT sessions, ECM, initial brain metastases, extracerebral progression at SRT1, a high BMV grade, no WBRT and local recurrence were also associated with a lower NDFS.

In contrast, relative to the four historical studies that compared SRT to SRT + WBRT, which found a median OS ranging from 8 to 15.2 months in the SRT group [5,9,20,21], we found a median OS of 18.6 months (95% CI 17.0–21.1). Kocher et al. found no significant difference in OS between patients receiving SRT and those undergoing WBRT [5]. In this randomized trial, 359 patients treated by complete surgery or radiosurgery were randomly assigned to adjuvant WBRT or observation. The characteristics of age, sex, ECOG, the primary cancer and the control of the primary of the patients included in this study were similar to the initial characteristics of the patients included in our analysis. In the observation group of the EORTC study, 33.5% of patients underwent salvage WBRT, whereas it was only 18.5% in our series. Salvage WBRT was mostly performed when the number of new or progressive BMs was high, the time since the last SRT session was short and the extracranial disease was not controlled. The median OS in patients treated with salvage WBRT in our series was 17.3 months (95% CI 16.2–21.1) after SRT1 and 8.8 months (95% CI 5.9–11.7) after WBRT compared to 10.9 months (95% CI 9.5- 14.2) and 11.6 months (95% CI 9.9–18.0) after WBRT in the series by Kocher and Brown, respectively [5,9]. Several previous studies including patients treated by more than two SRT sessions reported that the median OS ranged from 11 to 22.4 months [22,23,24,25]. Among them, Shuto et al. reported the highest median OS at 22.4 months, likely because only 16 patients treated with a high number of sessions (mean of 5.6 sessions) were analyzed [22]. Indeed, in our series, the median OS of the 12% of patients who underwent four or more SRTs was comparable at 57.4 months (95% CI 30.7-NA). With respect to high, intermediate and low BMV grades, Farris et al. reported the median OS times following the initial distant brain failure of 4.3, 8.4 and 12.4 months, respectively, while, as we found, the median OS times after SRT2 were 4.2, 10.4 and 12.5 months, respectively. We also found a significant association between the BMV grade and NDFS [16]. Farris et al. found that having more than two BMs at SRT1 was associated with a high BMV, while we found an association between the number of BMs at SRT1 and a higher BVM at SRT2 (*p* = 0.04), SRT3 (*p* = 0.02) and SRT4 (*p* = 0.008) and the final BMV (*p* < 0.001), but the number of BMs was not directly associated with either OS or NDFS and was a confounding factor for BMV.

None of the studies investigated repeated courses of SRT or analyzed the evolution of BM characteristics over time. BMs treated at SRT1 were larger, more numerous and more frequently operated on than BMs in subsequent SRTs [22,23,25,26,27,28,29]. This is in contrast to the BMs treated in SRT2 and following SRT which are less numerous and less voluminous due to systematic and close brain MRI monitoring [30].

The time between the two SRT sessions remained stable from one session to the next, whereas we might have expected an increase due to systemic disease control, systemic therapy and the diminution of the number of tumoral cells or a decrease due to an escape from radiotherapy or systemic therapy. The observed median time between the two SRT sessions on the current series is comparable to those reported by other authors. Thus, between the two sessions of SRT, Koiso et al. reported 6.4 months (min–max: 0.5–74.4) [26], Kim et al. reported 6.4 months (min–max: 2.5–42.7) for [31] and Fritz et al. reported 5.8 months (min–max: 0.9–35) [27].

The analysis of these series reporting results of repeated SRT sessions introduces an inherent selective survival bias. This study involved a very selective sub-group of patients with various primary cancers. Therefore, the question of whether patients survive longer because they are repeatedly treated or whether they are repeatedly treated because they survive longer remains. Kuntz et al. showed that patient characteristics were stable during all repeated SRT sessions, especially KPS, extracerebral progression, the BM volume, the number of BMs and surgical management [18]. The longer patients survive, the more likely they are to present local or distant brain relapse [32,33]. The results of this analysis reflect the patient condition more rather than a causal relationship between the number of SRT sessions and overall survival. This is why we have attempted to break this vicious cycle in two ways. First, we identified patients who were long-term survivors or responders to SRT. Patients surviving more than 2 years after SRT1 had a mean of three SRT sessions, a long time between two SRT sessions, a low BMV grade and no WBRT. Second, we studied survival after SRT1 or after SRT3 for patients with at least three SRT sessions. A high KPS at SRT1 and a low BMV grade at SRT2 were associated with better OS. Some previous studies have shown a controversial correlation between OS and primary cancer in patients treated for BM [25,34]. However, we identified an association between the primary cancer and OS, NDFS and long-term survival, suggesting that not all patients benefit equally from repeated SRT depending on their primary tumor.

Among patients with a poor prognosis (active extracranial disease, high BMV grade), the dilemma now lies between choosing the stereotactic treatment or opting for palliative care without a specific treatment. However, this choice can be particularly challenging for patients with poor prognostic outlooks, but who exhibit distressing symptoms that could potentially be alleviated through a single fraction of stereotactic treatment. In the era of personalized medicine, prognostic and predictive nomograms are increasingly used in oncology [35]. Nieder et al. developed prognostic and predictive nomograms of OS, NDFS, and new distant BMs in patients with BMs from solid tumors [36]. Age, sex, ECOG, KPS, histology of the primary tumor, tumor stage, primary site control, ECM, the number of BMs and the volume of the largest BMs were predictive of OS in the different nomograms. These results were very similar to the predictive factors for OS we found in the multivariate analysis. Holub et al. studied the predictive factors of the OS for 47 patients treated for 55 local recurrent metastases. The absence of ECM at diagnosis and during salvage SRT, as well as the absence of distant BM, were associated with better OS after SRT2 [37]. These data were also found to be predictive of OS in both the univariate and multivariate analyses in the current study. In 2010, Ammirati et al. showed that there were insufficient data to make clear recommendations for the management of local recurrent or distant BMs [38]. Today, salvage and repeated SRT has become an accepted standard of care in the treatment of a limited number of new BMs [39,40]. Nevertheless, the definition of limited brain diseases is constantly evolving, from one to three BMs a few years ago [41,42] to up to five [43,44] and sometimes up to ten if the cumulative volume is less than 25–30 mL [45,46,47]. Further retrospective and prospective comparative studies are needed to validate our findings. A nomogram based on patient characteristics at diagnosis and at each SRT could help clinicians identify which patients will benefit from salvage SRT.

## 5. Conclusions

Based on these results, patients who would benefit most from repeat SRT are those treated for kidney or breast cancer, those in good general condition, those who did not receive systemic treatment or ECM and those with a low BMV grade. Finally, a high KPS at SRT1 and a low BMV grade are prognostic factors for better OS, regardless of the number of BM recurrence events.

## Figures and Tables

**Figure 1 cancers-15-04948-f001:**
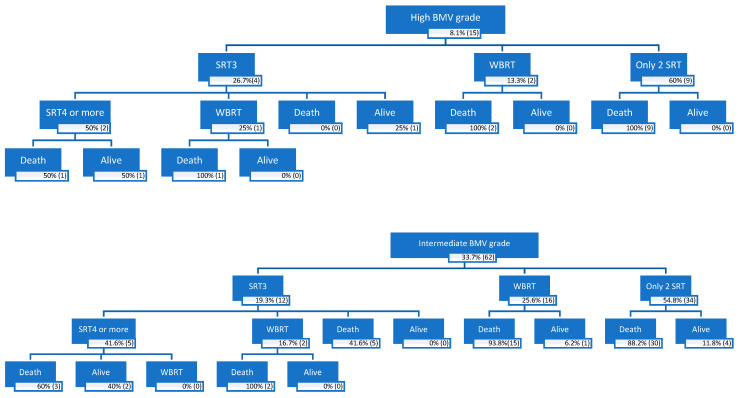
Patient outcome by BMV grade after SRT2. SRT2: Second Session of Stereotactic Radiotherapy; SR3: Third Session of Stereotactic Radiotherapy; SRT4: Forth Session of Stereotactic Radiotherapy.

**Figure 2 cancers-15-04948-f002:**
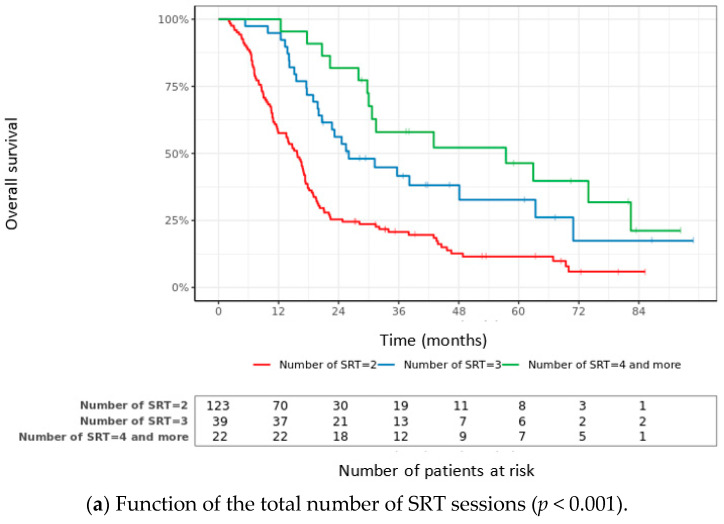
Overall survival. (**a**) Function of the total number of SRT sessions; (**b**) Function of BMV grade 3.5. Neurological death-free survival.

**Figure 3 cancers-15-04948-f003:**
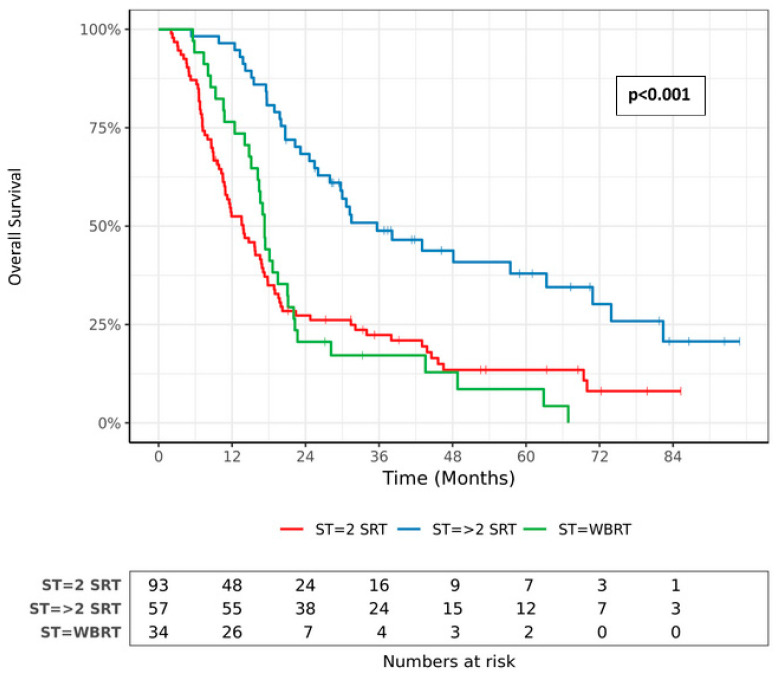
Overall survival according to salvage treatment. ST: Salvage Treatment; SRT: Stereotactic Radiotherapy; WBRT: Whole-Brain Radiotherapy.

**Table 1 cancers-15-04948-t001:** (**A**) Initial patient characteristics (*n* = 184). (**B**) Patient characteristics at each SRT.

(A)			
Characteristics		Number	Percentage
Sex	Male	91	49.5%
	Female	93	50.5%
Age at diagnosis of cancer	Median (range)	58 (21–87)	
	≤65 y	130	70.7%
	>65 y	54	29.3%
Medical history	Hypertension	75	40.8%
	Renal failure	12	6.5%
	Cholesterol	59	32.1%
	Smoking	118	64.1%
	Alcohol	26	14.1%
	Diabetes	17	9.2%
Primary cancer	Lung	103	56.0%
	Adenocarcinoma	70	70%
	Epidermoid	15	14.6%
	Small cell	4	3.9%
	Undifferentiated	4	3.9%
	Other	7	6.8%
	Breast	24	13.0%
	Luminal A	5	20.8%
	Luminal B	5	20.8%
	Her2 +	11	45.8%
	Triple negative	3	12.5%
	Melanoma	24	13.0%
	Kidney	8	4.4%
	Gastro-intestinal	16	8.7%
	Other	9	4.9%
Initial tumor stage (T)	1	23	12.5%
	2	47	25.5%
	3	38	20.7%
	4	30	16.3%
	Unknown	46	25.0%
Initial node stage (N)	0	37	20.1%
	1	23	12.5%
	2	49	26.6%
	3	26	14.1%
	Unknown	49	26.6%
Initial metastasis stage (M)	0	81	44.0%
	1	90	48.9%
	Of which synchronous BM	69	37.5%
	Unknown	13	7.1%
(**B**)			
**Characteristics at SRT1 (*n* = 184)**	Median age (range)		61 (24–88)
	Median ECOG (range)		1 (0–3)
	Median KPS (range)		90% (40–100)
	Median DS-GPA (range)		2.5 (0–4)
	Median RPA (range)		2 (1–3)
	Extracerebral progression		64%
	Control of the primary tumoral site		59.2%
	Systemic treatment		70%
	Median number of BMs (range)		1.5 (1–10)
	Median volume of BMs in mL (range)		0.4 (0.1–39.6)
	Postoperative tumor bed radiosurgery		38%
**Characteristics at SRT2 (*n* = 184)**	Median age (range)		61.5 (24.8–88.9)
	Median ECOG (range)		1 (0–3)
	Median KPS (range)		80% (50–100)
	Median DS-GPA (range)		2.5 (0–4)
	Median RPA (range)		2 (1–3)
	Extracerebral progression		30%
	Systemic treatment		69%
	Median number of BMs (range)		1 (1–9)
	Median volume of BMs in mL (range)		0.4 (0.1–59.7)
	Postoperative tumor bed radiosurgery		10%
	Reirradiation for local recurrent BM		15%
**Characteristics at SRT3 (*n* = 61)**	Median age (range)		60.6 (31.5–79.1)
	Median ECOG (range)		1 (0–2)
	Median KPS (range)		90% (60–100)
	Median DS-GPA (range)		2.5 (0–4)
	Median RPA (range)		2 (1–3)
	Extracerebral progression		23%
	Systemic treatment		77%
	Median number of BMs (range)		1 (1–8)
	Median volume of BMs in mL (range)		0.25 (0.1–48.6)
	Postoperative tumor bed radiosurgery		7%
	Reirradiation for local recurrent BM		16%

DS-GPA: Diagnosis-Specific Graded Prognostic Assessment; KPS: Karnofsky Performance Score; RPA: Recursive Partitioning Analysis; SRT1: First session of Stereotactic Radiotherapy; SRT2: Second session of Stereotactic Radiotherapy; SRT3: Third session of Stereotactic Radiotherapy.

**Table 2 cancers-15-04948-t002:** Overall survival and neurological death (*p* value).

Characteristics		Overall Survival		Neurological Death-Free Survival	
		U	M	U	M
Initial patient characteristics	Sex	<0.01	0.38	0.08	0.23
	Hypertension	0.05	0.18	0.3	
	Renal failure	0.3		0.8	
	Hyperlipidemia	0.5		0.4	
	Smoking	0.9		0.3	
	Alcohol	0.2		0.7	
	Diabetes	0.01	0.57	0.9	
Initial tumor characteristics	Primary tumor	0.02	<0.001	0.04	<0.001
	Tumor stage	0.02	0.24	0.03	<0.001
	Node stage	0.1		0.7	
	Metastatic stage	0.1		0.8	
	Synchronous BM	0.6	0.67	0.2	<0.01
	Extracerebral metastasis	0.08	0.05	0.6	<0.01
	Primary tumor control	0.3		0.6	
SRT	Number of SRT sessions	<0.01		0.1	
	Number of SRT sessions ≥ 3	<0.01	<0.001	0.03	0.02
	Number of BMs	0.6		0.7	
	Total BMs > 5	0.5		0.3	
	Total BMs > 10	0.1		0.03	0.99
	Total GTVs > 3 mL	0.3		0.03	0.99
	Total GTVs > 5 mL	0.8		0.2	
	Total GTVs > 10 mL	0.8		0.2	
	Total GTVs > 20 mL	0.9		0.6	
Patient characteristics at first SRT	WHO	<0.01	0.45	0.2	0.26
	KPS	<0.01	<0.01	0.3	0.02
	DS-GPA	0.3		0.3	
	RPA	0.01		0.6	
	Extracerebral progression	<0.01	0.81	0.3	<0.01
	Systemic treatment	0.04	0.01	0.8	0.08
	Number of BMs per patient	0.2		0.7	
	Symptom pre-SRT	0.6		0.8	
Worst/all patient characteristics	Worst DS-GPA	<0.01		0.1	
	Worst RPA	<0.01		0.2	
	Immunotherapy all time	0.8		0.8	
	Targeted therapy all time	0.02		0.1	
	Immunotherapy or targeted therapy all time	0.04		0.1	
	BMV grade	<0.01	<0.01	<0.01	<0.01
	Salvage WBRT	0.04	0.047	0.4	<0.01
	Local recurrence	<0.01	0.77	0.3	<0.01
	Radionecrosis	<0.01	0.42	0.6	<0.01

BMV: Brain Metastasis Velocity; DS-GPA: Diagnosis-Specific Graded Prognostic Assessment; GTV: Gross Tumor Volume; KPS: Karnofsky Performance Score; M: Multivariate Analysis; RPA: Recursive Partitioning Analysis; SRT: Stereotactic Radiotherapy; SRT1: First session of Stereotactic Radiotherapy; U: Univariate Analysis; WBRT: Whole-Brain Radiotherapy.

**Table 3 cancers-15-04948-t003:** Factors associated with OS regardless of the number of SRTs.

		Univariate	Multivariate
Initial patient characteristics	Sex	0.1	
	Initial brain metastasis	0.3	0.79
	Extracerebral metastasis	0.08	0.51
	Age at diagnosis of cancer > 65 y	0.007	0.36
	Age at diagnosis of brain metastasis > 65 y	0.3	
	Control of the primary tumor	0.1	
	Hypertension	0.7	
	Kidney disease	0.8	
	Hyperlipidemia	1	
	Tobacco	0.7	
	Alcohol	0.4	
	Diabetes	0.1	
	Primary cancer	0.08	0.5
	Initial tumor stage	0.03	0.25
	Initial node stage	0.07	
	Initial metastasis stage	0.08	
Patient characteristics at first SRT	Age > 65 y	0.3	
	WHO	<0.001	0.36
	KPS	<0.001	0.03
	DS-GPA	0.8	
	RPA	0.004	
	Extracerebral progression	0.007	0.58
	Systemic treatment	0.01	0.08
	Number of metastases per patient	0.5	
	Symptom pre-SRT	0.2	
Patient characteristics at second SRT	Age > 65 y	0.6	
	WHO	<0.001	0.52
	KPS	<0.001	0.3
	DS-GPA	0.4	
	RPA	<0.001	
	Extracerebral progression	0.05	0.1
	Systemic treatment	0.008	0.75
	Number of metastases per patient	0.4	
	Symptom pre-SRT	0.07	
	BMV Grade at SRT 2	<0.001	
	Low versus high		0.02
	Low versus intermediate		0.04

DS-GPA: Diagnosis-Specific Graded Prognostic Assessment; KPS: Karnofsky Performance Score; RPA: Recursive Partitioning Analysis; SRT1: First session of Stereotactic Radiotherapy; SRT2: Second session of Stereotactic Radiotherapy.

## Data Availability

Data available on request due to privacy restrictions. The data presented in this study are available on request from the corresponding author.

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
