# Peer review of "Repeated Stereotactic Radiotherapy for Local Brain Metastases Failure or Distant Brain Recurrent: A Retrospective Study of 184 Patients"

_cancers, 2023, doi:10.3390/cancers15204948_

Round 1
Reviewer 1 Report
In general: Well-structured and written article.
The topic is important; the patient series is worth the publication.
Nevertheless, I believe too much focus is placed on the prognosis of patients who received only 2 SRT sessions. The risk of a statistical coincidence of this result is considerable. As a bias, it can be postulated that the general condition of the patients became too bad to perform further brain stereotaxies. To investigate this suspicion, the following question arises:
- Can the group of patients who received only 2 SRT be compared with the other patients in terms of KPS and tumor extent? Are there any differences? It is striking that the median KPS score is only 80%, while it is higher in the other groups (cf. Table 1b).
Statistically, the statement is also risky since the patient groups become smaller and smaller as the number of stereotaxies increases!
In my view, the collective must be analyzed in this respect before a publication:
- the results section should be supplemented accordingly.
- The discussion should be expanded to include a section that clearly describes and discusses the study's limitations.
Minor corrections:
Table 2, "initial tumour characteristics", please define "primitive tumour" and "primitive tumour control", what does "primitive" mean?
The same for Table 3
Author Response
Dear Reviewer 1,
We would like to extend our sincere gratitude to the reviewer for their valuable comments, which have significantly contributed to improving the quality and readability of the manuscript. We have provided detailed responses to each of the reviewer's comments, addressing them point by point. We hope that the reviewer appreciates our responses and will consider granting their approval for the publication of this series analysis.
You'll find attached point-by-point answers to the reviewers' comments. Changes to the manuscript have been left visible.
Major corrections:
Reviewer:
In general: Well-structured and written article.
The topic is important; the patient series is worth the publication.
Nevertheless, I believe too much focus is placed on the prognosis of patients who received only 2 SRT sessions. The risk of a statistical coincidence of this result is considerable. As a bias, it can be postulated that the general condition of the patients became too bad to perform further brain stereotaxies. To investigate this suspicion, the following question arises:
- Can the group of patients who received only 2 SRT be compared with the other patients in terms of KPS and tumor extent? Are there any differences? It is striking that the median KPS score is only 80%, while it is higher in the other groups (cf. Table 1b).
Statistically, the statement is also risky since the patient groups become smaller and smaller as the number of stereotaxies increases!
In my view, the collective must be analyzed in this respect before a publication:
- the results section should be supplemented accordingly.
- The discussion should be expanded to include a section that clearly describes and discusses the study's limitations.
Reply:
Firstly, we would like to express our gratitude to the reviewer for their comments, which can enhance the quality of the manuscript.
Regarding the alteration of patient characteristics, a previous paper by the same authors, focusing on the same group of patients, has already investigated changes in patient characteristics throughout various SRT sessions and has been published. (DOI : 10.1186/s13014-023-02200-z)
A sentence and a corresponding reference have been included in paragraph 3.1: ' Further analyses of changes in patient characteristics over time are available in a dedicated article[18]. Repeated SRT does not lead to frank alteration of the general condition of the patients, KPS was maintained over 70% for more than 95% of our patients during all SRT.......' The discussion section has been expanded, and two additional article references have been included to provide a more comprehensive analysis and address the study's limitations.
Minor corrections:
Reviewer:
Table 2, "initial tumour characteristics", please define "primitive tumour" and "primitive tumour control", what does "primitive" mean?
The same for Table 3
Reply:
“Primitive” has been changed for “primary” in table 2 and table 3.
Reviewer 2 Report
Title: Overall survival and neurological death-free survival of patients treated with repeated stereotactic radiotherapy for locally recurrent brain metastases or distant brain failure: Can long- surviving patients be identified early? A retrospective study of 184 patients
In this manuscript, the authors ask whether long- surviving patients can be identified early?
To answer this question authors analyzed 184 patients treated with multiple stereotactic sessions
They reported that number of SRS session correlates with OS. Also, showed that KPS at initial diagnosis and BMV at second SRT correlates with survival . Another advantage of the study is that the authors reports that 37,5% of all patients survive more than 2 years after multiple sessions of SRT and median number of SRT sessions of long term survivors was 3.
Among short term surivors 27 patients receive WBRT , It would be of great interest if there was an OS difference between patients with intermediate and high BMV related to SRS or WBRT selected as a second session of RT.
In current form of manuscript it is not clear why authors excluded patients treated with single course of radiosurgery, this needs to be better emphasized by corrected title and purpose.
Also ability to assess a neurological death in retrospective study is very limiting. Therefore in current form I recommend this paper to major review. Although I support intention that longer survival may be achieved due to additional stereotactic radiosurgery. There are at least one additional analysis needed to confirm this conclusion.
Overall, this is an interesting study that provides some additional evidence to what is already known from the available literature. The manuscript seems valuable in principle, but I have a following comments:
1. Title – it is too long and complicated, please modify it and present clearly what have you studied –
Eg. Repeated stereotactic radiosurgery for brain metastases – analysis of 183 cases.
2. Conclusion – patients receiving multiple repeated stereotactic radiotherapy for locally recurrent brain metastases or distant brain failure have better OS and NDFS than those receiving only 2 SRT sessions.
It is a result not a conclusion, please remove.
3. Neurological death free survival:
- as the character of the study is retrospective, please explain how you define neurological cause of death
- section 3.6. – please clarify what are the favorable primary tumor sites
Other Methods: SRT was mostly delivered in one fraction of 20 Gy or in three fractions of 11 Gy every two days, with a 70% isodose line, 14 Gy or 23.1 Gy, respectively, encompassing the PTV. It is not a routine fashion of prescribing – please refere to other studies or guidelines
Results:
Please present an analysis of OS among patients with intermediate BMV treated with SRT or WBRT. This may improve our knowledge and be more informative what to do in such cases.
3.1 – why the primary sites are not summing up to 100%?’
3.2. What was the main reason that the patients with low BMV have not received further treatments – extent of intracranial or extracranial disease or both?
Discussion:
As you treated patients with high BMV and patients with active Extracranial disease I would recommend to discuss Stereotactic Radiosurgery of Patients with a Poor Prognosis -it is emerging but not a routine management.
Minor:
Line 71: Intercalated WBRT- what do you mean please explain
Line 285: Lung cancer, melanoma and other cancers were associated with a lower NDFS, 285
a high tumoral stage, a low number of SRTs, ECM, initial brain metastases, low KPS, ex- 286
tracerebral progression at SRT1, high BMV grade, no WBRT and local recurrence. – please rewrite , not clear.
included in the review file, overall quality is high
Author Response
Dear Reviewer 2,
We would like to extend our sincere gratitude to the reviewer for their valuable comments, which have significantly contributed to improving the quality and readability of the manuscript. We have provided detailed responses to each of the reviewer's comments, addressing them point by point. We hope that the reviewer appreciates our responses and will consider granting their approval for the publication of this series analysis.
You'll find attached point-by-point answers to the reviewers' comments. Changes to the manuscript have been left visible.
Major corrections:
Reviewer:
Title: Overall survival and neurological death-free survival of patients treated with repeated stereotactic radiotherapy for locally recurrent brain metastases or distant brain failure: Can long- surviving patients be identified early? A retrospective study of 184 patients
In this manuscript, the authors ask whether long- surviving patients can be identified early.
To answer this question authors analyzed 184 patients treated with multiple stereotactic sessions.
They reported that number of SRS session correlates with OS. Also, showed that KPS at initial diagnosis and BMV at second SRT correlates with survival. Another advantage of the study is that the authors reports that 37,5% of all patients survive more than 2 years after multiple sessions of SRT and median number of SRT sessions of long-term survivors was 3.
Among short term survivors 27 patients receive WBRT, It would be of great interest if there was an OS difference between patients with intermediate and high BMV related to SRS or WBRT selected as a second session of RT. In current form of manuscript, it is not clear why authors excluded patients treated with single course of radiosurgery, this needs to be better emphasized by corrected title and purpose.
Reply:
The reviewer's comment highlighted a significant issue regarding the clarity of the manuscript. We are deeply appreciative of the reviewer for providing us with the opportunity to enhance the text. Consequently, we wish to clarify that this article specifically focuses on repeated irradiation under stereotactic conditions. Therefore, patients who underwent whole-brain radiation therapy (WBRT) as their second cerebral treatment were not included in the current series. Likewise, patients who did not experience subsequent local or cerebral recurrence after their initial stereotactic session were also excluded.
Reviewer:
Also ability to assess a neurological death in retrospective study is very limiting. Therefore, in current form I recommend this paper to major review. Although I support intention that longer survival may be achieved due to additional stereotactic radiosurgery. There is at least one additional analysis needed to confirm this conclusion.
Overall, this is an interesting study that provides some additional evidence to what is already known from the available literature. The manuscript seems valuable in principle, but I have a following comments:
- Title – it is too long and complicated, please modify it and present clearly what have you studied – Eg. Repeated stereotactic radiosurgery for brain metastases – analysis of 183 cases.
Reply:
According to the relevant comment of the reviver, the title has been shortened for “Repeated stereotactic radiotherapy for locally brain metastases failure or distant brain recurrent: A retrospective study of 184 patients”
Reviewer:
- Conclusion – patients receiving multiple repeated stereotactic radiotherapy for locally recurrent brain metastases or distant brain failure have better OS and NDFS than those receiving only 2 SRT sessions. It is a result not a conclusion, please remove.
Reply :
We would like to express our appreciation to the reviewer for pointing out the confusion between the results and the actual conclusion. As a result, we have removed the problematic sentence from both the abstract's conclusion and the conclusion of the article.
Reviewer:
- Neurological death free survival: - as the character of the study is retrospective, please explain how you define neurological cause of death
Reply:
As pointed out by the reviewer, we acknowledge that the definition of neurological death was not clearly articulated. Therefore, we have added a sentence to clarify how we defined brain death: 'When the patient passed away at home or in another healthcare facility, the cause of death was predominantly recorded as unknown. In cases where the patient passed away in the hospital, death from a neurological cause was defined as either status epilepticus or brain involvement
Reviewer:
- section 3.6. – please clarify what are the favorable primary tumor sites
Reply:
In the multivariate analysis, favorable prognostic factors of NDFS were :
- breast cancer (digestive versus breast cancer: HR = 15.0, 95%CI 4.1–55.4, p < 0.001.
- lung versus breast cancer: HR = 9.64, 95%CI 4.0623.3, p < 0.001.
- melanoma versus breast cancer: HR = 5.59, 95%CI 1.7–18.1, p < 0.01.
- other cancers versus breast cancer: HR = 96.07, 95%CI 11.7–790.0, p < 0.001
And also :
- early tumoral stage (HR= 2.89, 95%CI 1.8–4.6, p< 0.001),
- no brain metastasis at primary cancer diagnostic (HR=0.31, 95%CI 0.1–0.9, p = 0.01),
- initial absence of ECM (HR = 0.17, 95%CI 0.07–0.44, p < 0.01),
- number of SRTs ≥ 3 (HR = 0.28, 95%CI 0.1–0.8, p = 0.02),
- high KPS at SRT1 (HR = 0.97, 95%CI 0.95–0.99, p = 0.02),
- no extracerebral progression at SRT1 (HR = 7.88, 95%CI 2.9–21.3, p < 0.01),
- BMV grade (low versus high: HR = 851.5; 95%CI 85.6–8468.7, p < 0.001;
- Low BMV versus intermediate: HR = 22.2; 95%CI 7.7–63.8, p < 0.01),
- no salvage WBRT (HR = 0.049; 95%CI 0.01–0.23, p < 0.001),
- no local recurrence (HR = 3.56; 95%CI 1.5–8.7, p < 0.01)
- Presence of radionecrosis (HR = 0.11, 95%CI 0.04–0.33, p < 0.01)
Reviewer:
Other Methods: SRT was mostly delivered in one fraction of 20 Gy or in three fractions of 11 Gy every two days, with a 70% isodose line, 14 Gy or 23.1 Gy, respectively, encompassing the PTV.
It is not a routine fashion of prescribing – please refere to other studies or guidelines.
Reply:
We have clarified the prescription method employed in Pitié-Salpêtrière Hospital (Paris) since 1994, in our center since 2010, and which is also utilized in several centers across the country. Specifically, a total dose of 33 Gy was prescribed to the isocenter, administered in 3 fractions of 11 Gy every two days. Additionally, a one-fraction protocol, delivering 20 Gy to the isocenter, was preferred when the lesion measured less than 1 cm. The treatment plans were designed such that the 70% isodose line encompassed the Planning Target Volume (PTV), corresponding to doses of 23.1 Gy or 14 Gy, respectively [15].
Reviewer:
Results: Please present an analysis of OS among patients with intermediate BMV treated with SRT or WBRT. This may improve our knowledge and be more informative what to do in such cases.
Reply:
We appreciate the reviewer's feedback. This series of patients has generated numerous analyses, and those related to Brain Metastases Volume (BMV) have already been published by the same authors. This article has been included in the references.
DOI : 10.1186/s13014-023-02200-z.
Reviewer:
3.1 – why the primary sites are not summing up to 100%?
Reply:
We greatly appreciate the reviewer's feedback, which highlighted readability issues. Consequently, we have made corrections to the tables: Histological subtypes for lung and breast cancers have been italicized and aligned to the right. Additionally, we have ensured that the sum of the percentages of primary cancer equals 100%, and the sums of the histological subtypes also equal 100%
Reviewer:
3.2. What was the main reason that the patients with low BMV have not received further treatments – extent of intracranial or extracranial disease or both?
Reply:
The reviewer raised a pertinent question that prompted a counterintuitive reflection. In fact, a low Brain Metastasis Volume (BMV) corresponds to fewer than 4 new brain metastases per year. Consequently, patients with low BMV received fewer new radiotherapy sessions than patients developing more than 4 new brain lesions per year. This low incidence results in a reduced treatment requirement, which might give the impression that these patients are receiving less treatment compared to those with high or intermediate BMV. This point is elaborated upon in paragraph 2.3, titled 'Brain Metastasis Velocity,' within the Materials and Methods section
Reviewer:
Discussion: As you treated patients with high BMV and patients with active Extracranial disease I would recommend to discuss Stereotactic Radiosurgery of Patients with a Poor Prognosis -it is emerging but not a routine management.
Reply:
We acknowledge the relevance of the reviewer's comment regarding the need to include a section on the indications for managing pour prognostic patients. In today's landscape, with the advent of short treatment courses for stereotactic treatment and its minimal short-term side effects (typically 1-3 fractions lasting 20 minutes), the comparison with Whole-Brain Radiation Therapy (WBRT) becomes less relevant. The dilemma now lies between choosing stereotactic treatment or opting for palliative care without specific treatment.
However, this choice can be particularly challenging for patients with poor prognostic outlooks but who exhibit distressing symptoms that could potentially be alleviated through a single fraction of stereotactic treatment. Therefore, the inclusion of this discussion point in decision-making boards may be essential. We have incorporated this aspect into our discussion.
Minor corrections:
Reviewer:
Line 71: Intercalated WBRT- what do you mean please explain
Reply:
“Intercalated” has been changed for “intercurrent”.
Reviewer:
Line 285: Lung cancer, melanoma and other cancers were associated with a lower NDFS,
a high tumoral stage, a low number of SRTs, ECM, initial brain metastases, low KPS, extracerebral progression at SRT1, high BMV grade, no WBRT and local recurrence. – please rewrite, not clear.
Reply:
The sentence has been rewritten:” Lung cancer andmelanoma were associated with a lower NDFS compared to breast cancer. Low KPS, high tumoral stage at diagnosis of the primary, a low number of SRT session, ECM, initial brain metastases, extracerebral progression at SRT1, high BMV grade, no WBRT and local recurrence. were also associated with a lower NDFS.